# The association between long-term opioid therapy and composite infection-related dental outcomes

Anne C. Black[1,2]*, Kirsha S. Gordon[1,2], James D. Dziura[3,4,5], Declan T. Barry[6,7], Stephen Crystal[8,9], E. Jennifer Edelman[2,5], Gretchen Gibson[10,11], Michelle Hamilton[12], Marianne Jurasic[13,14], Yu Li[15], Brandon D.L. Marshall[15], Melissa Skanderson[1], Katie Suda[16,17], William C. Becker[1,2]

1 VA Connecticut Healthcare System, West Haven, Connecticut, United States of America, 2 Department of Internal Medicine, Yale School of Medicine, New Haven, Connecticut United States of America, 3 Department of Emergency Medicine, Yale School of Medicine, New Haven, Connecticut United States of America, 4 Yale Center for Analytical Sciences, New Haven, Connecticut United States of America, 5 Yale School of Public Health, New Haven, Connecticut United States of America, 6 Department of Psychiatry, Yale School of Medicine, New Haven, Connecticut United States of America, 7 APT Foundation, New Haven, Connecticut United States of America, 8 Rutgers University, School of Social Work, New Brunswick, New Jersey United States of America, 9 Rutgers University, Institute for Health, Health Care Policy, and Aging Research, New Brunswick, New Jersey United States of America, 10 Kansas City University, College of Dental Medicine, Kansas City, Missouri United States of America, 11 VA Healthcare System of the Ozarks, Fort Smith, Arkansas United States of America, 12 VA Office of Dentistry, Washington DC, District of Columbia, United States of America, 13 Boston University Goldman School of Dental Medicine, Boston, Massachusetts United States of America, 14 VA Bedford Healthcare System, Bedford, Massachusetts United States of America, 15 Brown University School of Public Health, Providence, Rhode Island United States of America, 16 University of Pittsburgh, School of Medicine, Pittsburgh, Pennsylvania United States of America, 17 VA Pittsburgh Healthcare System, Pittsburgh, Pennsylvania United States of America

☉ These authors contributed equally to this work.
* anne.black@yale.edu

## Abstract

### Background

The Food and Drug Administration's warning that transmucosal buprenorphine, a partial opioid agonist used to treat opioid use disorder and chronic pain, may cause dental disease opens questions about potential class-wide adverse effects involving more widely prescribed opioid analgesics.

### Methods

This was a retrospective matched national cohort study of patients in care at the Department of Veterans Affairs (VA) between October 2010-September 2019. Patients prescribed LTOT were matched 1:2 to patients not prescribed LTOT on age, sex, service region, and VA dental coverage. Cox regression models estimated the association between LTOT and a composite infection-related dental outcome (CIDO). Sensitivity analyses excluded patients with cancer, restricted to patients with

**Data availability statement:** Data cannot be shared publicly because of security restrictions within the Department of Veterans Affairs, which require a data use and data transfer agreement when data are shared for research. Researchers may request access to a deidentified data set from the corresponding author. Researchers who meet the criteria for access to confidential data will be asked to sign a data use agreement. Data will be prepared for sharing in accordance with VA's data security guidelines. Contact information regarding data security: Privacy/FOIA Officer VA Connecticut Healthcare System 950 Campbell Avenue West Haven, CT 06516 CONPrivacyOfficers@va.gov.

**Funding:** The study was funded in part by the following: National Institute on Drug Abuse, 1RM1DA055310-01 to WCB. Website: https://nida.nih.gov/. Link to the specific grant: https://reporter.nih.gov/search/ykRq_72sLUux956Jeb-QmtQ/project-details/10652027. The funders played no role in study design, data collection or analysis or publishing decision. National Institute on Alcohol Abuse and Alcoholism, P01 AA029545-4 to ACJ (not an author). Website: https://niaaa.org/. Link to the specific grant: https://reporter.nih.gov/search/nEMK3R2Zl0K-dHeZJ5l4evA/project-details/10906187. The funders played no role in study design, data collection or analysis or publishing decision. National Institute on Alcohol Abuse and Alcoholism, U24 AA020794 to ACJ (not an author). Website: https://niaaa.org/. Link to the specific grant: https://reporter.nih.gov/search/5j6Au2pzsU2wWc_DYZozFA/project-details/10003103. The funders played no role in study design, data collection or analysis or publishing decision. National Institute on Alcohol Abuse and Alcoholism, U10 AA013566 to ACJ (not an author). Website: https://niaaa.org/. Link to the specific grant: https://reporter.nih.gov/search/rJptGFAHH0yEMvvIAHKwkw/project-details/7682335. The funders played no role in study design, data collection or analysis or publishing decision.

**Competing interests:** The authors have declared that no competing interests exist.

comprehensive dental coverage, and to patients with ≥180 days of follow-up time, respectively.

## Results

The cohort comprised 2,173,435 patients including 787,825 (36%) receiving LTOT; 612,101 (28%) experienced CIDO. In both simple and multivariable regression models, LTOT exposure was associated with greater CIDO risk; HR (95% CI) =1.24 (1.23, 1.25); aHR (95% CI) =1.08 (1.07, 1.08), respectively; p < 0.001. Sensitivity analyses showed similar results except among patients with full dental coverage for whom CIDO rates were substantially higher and LTOT was not statistically significantly associated with risk.

## Conclusions

he observed positive association between LTOT and CIDO in this large VA sample may inform patient-provider discussions and decision-making around use of LTOT. High CIDO rates among patients with full VA dental coverage may reflect their unique vulnerability to dental infection associated with service-related dental or disabling conditions. Limitations include risk for ascertainment bias, unclear generalizability to a broader clinical population, and the potential for residual confounding.

## Introduction

Clinical guidelines for chronic pain management de-emphasize use of long-term full-opioid agonist therapy (LTOT), such as oxycodone and hydrocodone, given risks for opioid-related harms including opioid use disorder (OUD) and equivocal evidence for long-term effectiveness [1,2]. For individuals prescribed LTOT with evidence of OUD, guidelines recommend transition to buprenorphine [3], a partial opioid agonist with analgesic effects and well-established effectiveness for OUD management. Transition to buprenorphine is preferred to drastic reduction or discontinuation of LTOT given the associated high risk for overdose and death [4]. However, in 2022, the Food and Drug Administration [5] published a warning about dental problems associated with transmucosal buprenorphine (e.g., Suboxone, a sublingual buprenorphine/naloxone film), the commonly prescribed buprenorphine formulation, citing risks of tooth decay, cavities, oral infections, and tooth loss. Concern over these risks may present a barrier to buprenorphine initiation in patients prescribed LTOT for whom such treatment is indicated. However, full opioid agonists themselves may pose oral health risks due to immunosuppression and well-documented effects on saliva flow causing xerostomia; both create opportunity for oral disease development [6,7]. Nevertheless, to our knowledge, no published studies have specifically addressed the dental risks of LTOT. The goal of the current study was to assess the effect of LTOT exposure on dental disease development in a retrospective cohort of patients receiving care within the Department of Veterans Affairs (VA).

## Methods

This study was conducted under waiver of informed consent with approval by the VA Connecticut Healthcare System Research and Development Committee and Yale University Human Investigation Committee. Data were accessed for this study between January 24, 2024-March 28, 2025. Authors with IRB-approved access to the data (ACB, KSG, MS, WCB) had access to information that could identify individual participants during and after the data access period.

Using a national cohort curated by the Veterans Aging Cohort Study (VACS), we created a matched cohort of patients who received VA primary care between October 1, 2010-September 30, 2019 (i.e., fiscal years [FY] 2011–2019) and had not experienced the primary outcome in the year prior to cohort entry. The dataset included all administrative, pharmacy, and healthcare data in VA electronic health records (EHR) for the study period. LTOT prescription was defined as ≥ 90 consecutive days, allowing for 30-day refill gaps, of hydrocodone, oxycodone, morphine, fentanyl, hydromorphone, dihydrocodeine, meperidine, pentazocine, propoxyphene, levorphanol, tramadol, or tapentadol (excluding methadone) [8]. Patients with incident LTOT exposure, with no exposure to LTOT in the prior year were matched 1:2 to patients not prescribed LTOT. Matching variables included age (in 5-year increments), sex, VA service region, and category of VA dental coverage (i.e., unknown/missing, no, some, or comprehensive coverage). The index date for LTOT-exposed patients was the day of LTOT initiation, and for those not LTOT-exposed, was the first VA visit date within the same FY as matched LTOT-exposed patients. The baseline period was 12 months prior to index date, during which patient characteristics were ascertained.

The primary outcome was the composite infection-related dental outcome (CIDO), a dichotomous indicator of any new (i.e., post-baseline) diagnosis of dental caries, oral infections, or tooth loss according to International Classification of Diseases (ICD) and Current Procedural Terminology codes adjudicated by dental expert coauthors (GG, MH, MJ, KS; Table in S1 Table).

We calculated descriptive statistics for the cohort overall and by LTOT exposure and outcome. Group differences were assessed by chi-square, t-tests, or Wilcoxon signed-rank tests. We used Cox proportional hazards modeling, accounting for matching, to estimate the unconditional effect of LTOT exposure on days to first recorded CIDO, death, or end of study. We then estimated a multivariable Cox model, adjusting for age at index date and baseline-recorded sex, race/ethnicity (Black, Hispanic, White, or other), smoking status (never, unknown, current, past), rurality of home address (rural/highly rural vs. not), pain level (none, non-chronic, or chronic, based on pain-related diagnosis), serious mental health condition (major depressive disorder, post-traumatic stress disorder, bipolar disorder, schizophrenia), medical conditions (cancer, chronic obstructive pulmonary disease, stroke, diabetes, human immunodeficiency virus [HIV], hepatitis C [HCV], cardiovascular disease, liver disease), substance use (alcohol use disorder, non-opioid drug use disorder, opioid use disorder), obesity (body mass index ≥30) [9], and overall illness severity (measured by the VACS index) [10]. Patients with incomplete data and those who died during the baseline period were excluded. Pain, mental health, and medical conditions were determined by ICD codes.

We conducted several sensitivity analyses to test for stability in model estimates: (1) excluding patients with cancer who may be uniquely vulnerable to infection, (2) restricting to patients with comprehensive VA dental coverage provided for service-connected dental disability or other disabling medical condition [11,12] who may be particularly vulnerable to infection and whose dental care records may be differentially available in the VA EHR, and (3) restricting to patients with ≥180 days of follow-up time to standardize the time at risk for CIDO.

## Results

The cohort, comprising 2,173,435 patients, was predominantly male (92%) and White (67%), consistent with the VA patient population. The majority smoked or had smoked (69%), 38% lived in a rural setting, and approximately half had obesity (51%) and chronic pain (46%; Table 1). Consistent with the 1:2 matching schema, approximately one-third (36%) were prescribed LTOT. A total of 28% experienced CIDO.

**Table 1. Patients in care FY 2010-2019 characteristics, n = 2,173,435.**

| | Overall | Exposure | | Outcome | |
|---|---|---|---|---|---|
| | Mean (SD) or N (%) | LTOT, n = 787825 | No LTOT, n = 1385610 | CIDO, n = 612101 | no CIDO, n = 1561334 |
| **Age**, mean (SD) | 58 (13) | 57 (13) | 58 (13)** | 56 (12) | 58 (13)** |
| **Sex on record** | | | | | |
| Female | 162172 (7.46) | 60352 (7.66) | 101820 (7.35)** | 51145 (8.36) | 111027 (7.11)** |
| Male | 2011263 (92.54) | 727473 (92.34) | 1283790 (92.65)** | 560956 (91.64) | 1450307 (92.89)** |
| **Race/Ethnicity** | | | | | |
| Black | 413119 (19.01) | 132543 (16.82) | 280576 (20.25)** | 149641 (24.45) | 263478 (16.88)** |
| Hispanic | 140737 (6.48) | 45824 (5.82) | 94913 (6.85)** | 45813 (7.48) | 94924 (6.08)** |
| Other | 158499 (7.29) | 50880 (6.46) | 107619 (7.77)** | 40097 (6.55) | 118402 (7.58)** |
| White | 1461080 (67.22) | 558578 (70.90) | 902502 (65.13)** | 376550 (61.52) | 1084530 (69.46)** |
| **Smoking status** | | | | | |
| Never | 662620 (30.49) | 191098 (24.26) | 471522 (34.03)** | 178186 (29.11) | 484434 (31.03)** |
| Current | 786588 (36.19) | 357329 (45.36) | 429259 (30.98)** | 249407 (40.75) | 537181 (34.41)** |
| Past | 718448 (33.06) | 238132 (30.23) | 480316 (34.66)** | 183800 (30.03) | 534648 (34.24)** |
| Unknown/missing | 5779 (0.27) | 1266 (0.16) | 4513 (0.33)** | 708 (0.12) | 5071 (0.32)** |
| **Urban/Rural** | | | | | |
| Urban | 1355513 (62.37) | 464232 (58.93) | 891281 (64.32)** | 401671 (65.62) | 953842 (61.09)** |
| Rural | 784145 (36.08) | 309918 (39.34) | 474227 (34.23)** | 202075 (33.01) | 582070 (37.28)** |
| Highly rural | 30133 (1.39) | 12294 (1.56) | 17839 (1.29)** | 7474 (1.22) | 22659 (1.45)** |
| Unknown/missing | 3644 (0.27) | 1381 (0.18) | 2263 (0.16)** | 881 (0.14) | 2763 (0.18)** |
| **Pain level** | | | | | |
| None | 1094435 (50.36) | 196020 (24.88) | 898415 (64.84)** | 253741 (41.45) | 840694 (53.84)** |
| Non chronic | 68301 (3.14) | 22411 (2.84) | 45890 (3.31)** | 20244 (3.31) | 48057 (3.08)** |
| Chronic | 1010699 (46.50) | 569394 (72.27) | 441305 (31.85)** | 338116 (55.24) | 672583 (43.08)** |
| **Mental health conditions** | | | | | |
| Bipolar disorder | 58486 (2.69) | 29168 (3.70) | 29318 (2.12)** | 27525 (4.50) | 30961 (1.98)** |
| Major depression | 168053 (7.73) | 91634 (11.63) | 76419 (5.52)** | 71579 (11.69) | 96474 (6.18)** |
| PTSD | 289356 (13.31) | 139683 (17.73) | 149673 (10.80)** | 147119 (24.04) | 142237 (9.11)** |
| Schizophrenia | 36505 (1.68) | 11442 (1.45) | 25063 (1.81)** | 20287 (3.31) | 16218 (1.04)** |
| SMH (any of the above) | 449048 (20.66) | 217078 (27.55) | 231970 (16.74)** | 211432 (34.54) | 237616 (15.22)** |
| Stroke | 32758 (1.51) | 15937 (2.02) | 16821 (1.21)** | 11106 (1.81) | 21652 (1.39)** |
| COPD | 149661 (6.89) | 86921 (11.03) | 62740 (4.53)** | 47862 (7.82) | 101799 (6.52)** |
| Diabetes | 476855 (21.94) | 207014 (26.28) | 269841 (19.47)** | 155648 (25.43) | 321207 (20.57)** |
| HCV | 66863 (3.08) | 36332 (4.61) | 30531 (2.20)** | 26993 (4.41) | 39870 (2.55)** |
| HIV | 11837 (0.54) | 4426 (0.56) | 7411 (0.53)* | 4955 (0.81) | 6882 (0.44)** |
| Cancer | 122236 (5.62) | 60717 (7.71) | 61519 (4.44)** | 43051 (7.03) | 79185 (5.07)** |
| CVD | 64301 (2.96) | 31805 (4.04) | 32496 (2.35)** | 21290 (3.48) | 43011 (2.75)** |
| Liver disease | 63059 (2.90) | 34230 (4.34) | 28829 (2.08)** | 24051 (3.93) | 39008 (2.50)** |
| AUD | 160406 (7.38) | 72381 (9.19) | 88025 (6.35)** | 70794 (11.57) | 89612 (5.74)** |
| Drug use disorder | 109163 (5.02) | 53906 (6.84) | 55257 (3.99)** | 54854 (8.96) | 54309 (3.48)** |
| OUD | 20501 (0.94) | 12219 (1.55) | 8282 (0.60)** | 10358 (1.69) | 10143 (0.65)** |
| Obesity (BMI ≥ 30) | 1102421 (50.72) | 416598 (52.88) | 685823 (49.50)** | 321561 (52.53) | 780860 (50.01)** |
| Severity of illness, median (IQR) | 32 (26, 40) | 34 (27, 42) | 32 (26, 39)** | 32 (26, 39) | 33 (26, 40)** |

*(Continued)*

**Table 1.** (Continued)

| | Overall | Exposure | | Outcome | |
|---|---|---|---|---|---|
| | Mean (SD) or N (%) | LTOT, n=787825 | No LTOT, n=1385610 | CIDO, n=612101 | no CIDO, n=1561334 |
| **Dental class** | | | | | |
| Unknown/missing | 1231043 (56.64) | 458060 (58.14) | 772983 (55.79)** | 57401 (9.38) | 1173642 (75.17)** |
| No coverage | 619405 (28.50) | 218089 (27.68) | 401316 (28.96)** | 382896 (62.55) | 236509 (15.15)** |
| Some coverage | 315745 (14.53) | 109095 (13.85) | 206650 (14.91)** | 167013 (27.29) | 148732 (9.53)** |
| Comprehensive coverage | 7242 (0.33) | 2581 (0.33) | 4661 (0.34)** | 4791 (0.78) | 2451 (0.16)** |
| **LTOT** | 787825 (36.25) | | | 232944 (38.06) | 554881** (35.54) |
| **CIDO** | 612101 (28.16) | 232944 (29.57) | 379157 (27.36)** | | |
| **Died** | 542487 (24.96) | 239196 (30.36) | 303291 (21.89)** | 155769 (25.45) | 386718 (24.77)** |
| **FU**, median (IQR) | 7 (3, 9) | 6 (3, 9) | 7 (3, 9)** | 3 (1, 5) | 8 (5, 10)** |

PTSD = post-traumatic stress disorder, SMH = severe mental health condition, COPD = chronic obstructive pulmonary disease, HCV = hepatitis C, CVD = cardiovascular disease, AUD = alcohol use disorder, OUD = opioid use disorder, BMI = body mass index, LTOT = long-term opioid therapy, CIDO = composite of infection-related dental outcomes, FU = months of follow-up after index, **p < .01, *p < .05

Relative to patients not prescribed LTOT, those prescribed LTOT were more likely to be White, smoke currently, live in a rural setting, have chronic pain, a serious mental health condition (except schizophrenia), substance use disorder, greater illness severity, and experience CIDO. Although statistically significant, differences in VA dental coverage were negligible due to matching (Table 1).

Patients with CIDO, compared to those without, were more likely to be female, Black, smoke currently, reside in a non-rural setting, have chronic pain, a serious mental health condition, substance use disorder, VA dental coverage, and receive LTOT.

In the simple Cox model, being prescribed LTOT was associated with greater risk of CIDO compared to having no prescribed LTOT; hazard ratio (HR) (95% CI) = 1.24 (1.23, 1.25). Adjusting for covariates, the risk associated with LTOT was attenuated but remained statistically significant; adjusted hazard ratio (aHR) (95% CI) = 1.08 (1.07, 1.08).

Covariates associated with greater risk for CIDO included having a pain-related diagnosis, serious mental health condition, cancer diagnosis, substance use disorder, and higher illness severity. In contrast, increased age, White or other race (compared to Black), rural residence, HCV, and HIV were associated with lower CIDO risk (Table 2).

In individual sensitivity analyses limited to patients without cancer, and to those with at least 180 days of follow-up, respectively, results essentially replicated the multivariable full-cohort model's; aHR (95% CI) = 1.08 (1.07, 1.09) (Table 3). In contrast, among patients with comprehensive VA dental coverage, whose CIDO rates were substantially higher than among the cohort overall, there was no association between CIDO and LTOT; aHR (95% CI) = 0.97 (0.89, 1.05).

## Discussion

To our knowledge, this is the first study to demonstrate the association between LTOT exposure and dental disease. This finding is important in light of recent warnings of buprenorphine risks that may influence decision-making in the context of chronic pain and/or OUD. Evidence of LTOT-associated dental risks may inform clinical guidelines for LTOT prescribing, including recommendations for specific monitoring of patients' oral health, attention to oral healthcare while taking LTOT, and patient-provider discussions of dental risks before initiating LTOT. However, several study limitations should be considered when interpreting study results, including the risk for confounding by indication. Specifically, despite our effort to control for covariates of LTOT prescribing including substance use and medical comorbidities, unmeasured confounders may have driven observed associations attributed to LTOT. Secondly, the study was limited by our use of available data in patients' EHR. Patients without VA dental coverage may have CIDO recorded in other

**Table 2. Multivariable Cox proportional hazards model of CIDO, n = 2,173,435.**

| | aHR (95%) |
|---|---|
| **LTOT** | 1.08 (1.07, 1.08)** |
| **Age/10** | 0.95 (0.92, 0.97)** |
| **Race/Ethnicity** (ref Black) | |
| Hispanic | 1.01 (0.99, 1.02) |
| Other | 0.93 (0.92, 0.95)** |
| White | 0.98 (0.97, 0.99)** |
| **Smoking** (ref never) | |
| Current | 1.03 (1.02, 1.04)** |
| Past | 1.00 (0.99, 1.01) |
| Unknown/missing | 1.33 (1.18, 1.50)** |
| **Rurality** | 0.98 (0.97, 0.99)** |
| **Pain level** (ref none) | |
| Non chronic | 1.17 (1.16, 1.18)** |
| Chronic | 1.11 (1.09, 1.14)** |
| **SMH** | 1.43 (1.41, 1.44)** |
| **Stroke** | 1.00 (0.96, 1.04) |
| **COPD** | 1.05 (1.03, 1.07)** |
| **Diabetes** | 1.08 (1.07, 1.09)** |
| **HCV** | 0.93 (0.91, 0.96)** |
| **HIV** | 0.95 (0.91, 0.99)* |
| **Cancer** | 1.31 (1.29, 1.34)** |
| **CVD** | 1.07 (1.03, 1.10)** |
| **Liver disease** | 1.01 (0.99, 1.04) |
| **AUD** | 1.09 (1.07, 1.10)** |
| **Drug use disorder** | 1.20 (1.18, 1.22)** |
| **OUD** | 1.01 (0.98, 1.05) |
| **Obesity** (BMI ≥ 30) | 1.02 (1.01, 1.02)** |
| **Severity of illness/10** | 1.12 (1.12, 1.13)** |

aHR = adjusted hazard ratio, SMH = severe mental health condition (composite of major depression, PTSD, bipolar disorder, schizophrenia), COPD = chronic obstructive pulmonary disease, HCV = hepatitis C, CVD = cardiovascular disease, AUD = alcohol use disorder, OUD = opioid use disorder, BMI = body mass index, **$p < .01$, *$p < .05$.

systems not reviewed in the current study, posing risk for access-related ascertainment bias. However, given our matching paradigm, outcome misclassification likely would not differ by LTOT status. Whereas in this study we controlled for substance use disorders, future research is needed to account for the unique oral health-related risks of specific substance use in the context of LTOT exposure. Further, as conditions comprising CIDO (e.g., oral infection, tooth loss) differ in etiology and clinical course, our use of a composite outcome may mask mechanism-specific associations. Thus, future research on this topic should assess effects of LTOT on individual CIDO conditions. Sensitivity analyses suggested LTOT may pose minimal additional risk to patients already vulnerable to dental infection, such as patients provided VA comprehensive dental coverage. However, smaller sample size in that analysis may have masked differences in CIDO rate. Finally, given the study's focus on a patient sample receiving services within the VA, results may not generalize to a broader population of patients. Despite these limitations, the demonstrated association between

**Table 3. Sensitivity analyses: Model 1 excluded those with cancer, Model 2 restricted to those with comprehensive dental coverage, Model 3 restricted to those with index date ≥180 days prior to end of study.**

|  | Model 1 n=2,051,199 | Model 2 n=7242 | Model 3 n=2,156,900 |
|---|---|---|---|
|  | aHR (95%) | aHR (95%) | aHR (95%) |
| **LTOT** | 1.08 (1.07, 1.09)** | 0.97 (0.89, 1.05) | 1.08 (1.07, 1.08)** |
| **Age/10** | 0.94 (0.92, 0.97)** | 0.83 (0.60, 1.13) | 0.95 (0.92, 0.97)** |
| **Race/Ethnicity** (ref Black) |  |  |  |
| Hispanic | 1.00 (0.99, 1.02) | 0.95 (0.78, 1.15) | 1.01 (0.99, 1.02) |
| Other | 0.93 (0.91, 0.94)** | 0.88 (0.72, 1.06) | 0.93 (0.92, 0.95)** |
| White | 0.97 (0.96, 0.98)** | 0.91 (0.81, 1.03) | 0.98 (0.97, 0.99)** |
| **Smoking** (ref never) |  |  |  |
| Current | 1.03 (1.02, 1.05)** | 0.88 (0.79, 0.98)* | 1.03 (1.02, 1.04)** |
| Past | 1.00 (0.99, 1.01) | 0.93 (0.83, 1.04) | 1.00 (0.99, 1.01) |
| Unknown/missing | 1.31 (1.15, 1.51)** | 1.40 (0.30, 6.46) | 1.33 (1.18, 1.50)** |
| **Rurality** | 0.98 (0.97, 0.99)** | 0.98 (0.89, 1.08) | 0.98 (0.97, 0.99)** |
| **Pain level** (ref none) |  |  |  |
| Non chronic | 1.17 (1.16, 1.18)** | 1.21 (1.10, 1.33)** | 1.17 (1.16, 1.18)** |
| Chronic | 1.11 (1.09, 1.14)** | 1.22 (0.93, 1.59) | 1.11 (1.09, 1.14)** |
| **SMH** | 1.43 (1.42, 1.45)** | 1.22 (1.10, 1.36)** | 1.43 (1.41, 1.44)** |
| **Stroke** | 0.98 (0.94, 1.03) | 1.82 (0.88, 3.78) | 1.00 (0.96, 1.04) |
| **COPD** | 1.05 (1.03, 1.07)** | 1.16 (0.93, 1.46) | 1.05 (1.03, 1.07)** |
| **Diabetes** | 1.09 (1.08, 1.10)** | 0.97 (0.85, 1.11) | 1.08 (1.07, 1.09)** |
| **HCV** | 0.94 (0.91, 0.97)** | 1.05 (0.74, 1.50) | 0.93 (0.91, 0.96)** |
| **HIV** | 0.97 (0.92, 1.01) | 1.02 (0.52, 2.01) | 0.95 (0.91, 0.99)* |
| **Cancer** |  | 1.16 (0.92, 1.45) | 1.31 (1.29, 1.33)** |
| **CVD** | 1.10 (1.06, 1.14)** | 0.66 (0.40, 1.07) | 1.07 (1.03, 1.10)** |
| **Liver disease** | 1.02 (0.99, 1.04) | 0.93 (0.65, 1.33) | 1.01 (0.99, 1.04) |
| **AUD** | 1.09 (1.07, 1.10)** | 1.11 (0.90, 1.37) | 1.09 (1.07, 1.10)** |
| **Drug use disorder** | 1.20 (1.18, 1.23)** | 1.03 (0.78, 1.35) | 1.20 (1.18, 1.22)** |
| **OUD** | 1.02 (0.98, 1.06) | 0.80 (0.47, 1.35) | 1.01 (0.98, 1.05) |
| **Obesity** (BMI ≥ 30) | 1.02 (1.01, 1.03)** | 0.92 (0.84, 1.01) | 1.02 (1.01, 1.03)** |
| **Severity of illness/10** | 1.12 (1.11, 1.13)** | 1.00 (0.93, 1.07) | 1.12 (1.12, 1.13)** |

aHR=adjusted hazard ratio, SMH=severe mental health condition (composite of major depression, PTSD, bipolar disorder, schizophrenia), COPD=chronic obstructive pulmonary disease, HCV=hepatitis C, CVD=cardiovascular disease, AUD=alcohol use disorder, OUD=opioid use disorder, BMI=body mass index, **p<.01, *p<.05.

incident LTOT exposure and new diagnosis of dental disease in this rigorous, large-sample analysis exposes the need for additional study. To inform the critical decision to transition from LTOT to buprenorphine to address OUD, research is needed to assess the relative oral health risks of these medications.

## Supporting information

**S1 Table. Current dental terminology (CDT) and international classification of diseases (ICD) codes for the composite of infection-related dental outcomes (CIDO), a composite of dental caries, oral infections, and loss of teeth.**
(DOCX)

## Acknowledgments

This work uses data provided by patients and collected by the Department of Veterans Affairs as part of their care and support. The views and opinions expressed in this manuscript are those of the authors and do not necessarily represent those of the Department of Veterans Affairs or the United States government.

## Author contributions

**Conceptualization:** Anne C. Black, Kirsha S. Gordon, Stephen Crystal, E. Jennifer Edelman, Gretchen Gibson, Michelle Hamilton, Marianne Jurasic, Yu Li, Brandon DL Marshall, Melissa Skanderson, Katie Suda, William C. Becker.

**Data curation:** Kirsha S. Gordon, Melissa Skanderson.

**Formal analysis:** Kirsha S. Gordon, Melissa Skanderson.

**Funding acquisition:** Kirsha S. Gordon, Declan T. Barry, E. Jennifer Edelman, William C. Becker.

**Investigation:** Anne C. Black, Kirsha S. Gordon, James D. Dziura, Declan T. Barry, Stephen Crystal, E. Jennifer Edelman, Gretchen Gibson, Michelle Hamilton, Marianne Jurasic, Yu Li, Brandon DL Marshall, Melissa Skanderson, Katie Suda, William C. Becker.

**Methodology:** Anne C. Black, Kirsha S. Gordon, James D. Dziura, Stephen Crystal, E. Jennifer Edelman, Gretchen Gibson, Michelle Hamilton, Marianne Jurasic, Yu Li, Brandon DL Marshall, Melissa Skanderson, Katie Suda, William C. Becker.

**Project administration:** Declan T. Barry, William C. Becker.

**Resources:** Declan T. Barry, William C. Becker.

**Software:** Kirsha S. Gordon, Melissa Skanderson.

**Supervision:** James D. Dziura, Gretchen Gibson, William C. Becker.

**Validation:** Anne C. Black, Kirsha S. Gordon, James D. Dziura, Declan T. Barry, Stephen Crystal, E. Jennifer Edelman, Gretchen Gibson, Michelle Hamilton, Marianne Jurasic, Yu Li, Brandon DL Marshall, Melissa Skanderson, Katie Suda, William C. Becker.

**Visualization:** Anne C. Black, Kirsha S. Gordon, James D. Dziura, Declan T. Barry, Stephen Crystal, E. Jennifer Edelman, Gretchen Gibson, Michelle Hamilton, Marianne Jurasic, Yu Li, Brandon DL Marshall, Melissa Skanderson, Katie Suda, William C. Becker.

**Writing – original draft:** Anne C. Black, Kirsha S. Gordon.

**Writing – review & editing:** James D. Dziura, Declan T. Barry, Stephen Crystal, E. Jennifer Edelman, Gretchen Gibson, Michelle Hamilton, Marianne Jurasic, Yu Li, Brandon DL Marshall, Melissa Skanderson, Katie Suda, William C. Becker.

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
