## [Decision Letter · Decision Letter 0]

11 Nov 2025

PONE-D-25-46805The association of long-term opioid therapy and dental diseasePLOS ONE

Dear Dr. Black,

Thank you for submitting your manuscript to PLOS ONE. After careful consideration, we feel that it has merit but does not fully meet PLOS ONE’s publication criteria as it currently stands. Therefore, we invite you to submit a revised version of the manuscript that addresses the points raised during the review process. This article does bring attention to an important health outcome among chronic opioid users.  Both reviewers, and I agree, that significant information is needed in the Methods and Discussion sections. Please consider how bias may impact measures of excess risk.

We look forward to receiving your revised manuscript.

Kind regards,

Kimberly Page, PhD, MPH

Academic Editor

PLOS ONE

Journal Requirements:

Reviewers' comments:

Reviewer's Responses to Questions

**Comments to the Author**

1. Is the manuscript technically sound, and do the data support the conclusions?

Reviewer #1: Yes

Reviewer #2: Partly

2. Has the statistical analysis been performed appropriately and rigorously? 

Reviewer #1: Yes

Reviewer #2: Yes

3. Have the authors made all data underlying the findings in their manuscript fully available?

Reviewer #1: Yes

Reviewer #2: Yes

4. Is the manuscript presented in an intelligible fashion and written in standard English?

Reviewer #1: Yes

Reviewer #2: Yes

5. Review Comments to the Author

Reviewer #1: 1. Recommend changing title to “The association of long-term opioid therapy and Composite Infection-related Dental Outcomes”

2. In Methods, you state primary outcome related to new-onset dental caries. How was this determined given that caries is a chronic bacterial infection that may be present before subjective reports by patients or clinical/radiographic findings? Please define term “oral infections” to assist readership in knowing if this is referring to periodontal/endodontic and/or post-surgical infections. Was tooth loss inclusive to oral infections and exclusive of orthodontic and/or prophylactic removal of third molars?

3. In Methods, you mention you adjusted Cox model for substance use (drug use disorder, alcohol use disorder)

a. Type of drugs misused may be a confounding variable in this study as methamphetamine and opioids alter salivary function; methamphetamine decreases salivary pH

4. In Methods, please describe why patients with cancer, comprehensive dental coverage, and less than 180 days of follow-up were excluded? Please define the requirements for veterans to obtain comprehensive dental coverage? Are they more susceptible to long-term opioid exposure? Why?

5. Tables need to be organized with data presented in order mentioned in text (race, smoking status, urban/rural, pain, MH, CV, SUD etc.), Too much information provided in tables. Recommend increase the number of tables.

6. Results:

a. State patients prescribed LTOT live in rural settings-data shows 58.93% urban vs. 39.34% rural with more living in urban settings-please explain and/or correct your result statement

7. Discussion

a. Why are patients with VA comprehensive dental coverage already extremely vulnerable to dental infection? Please define this coverage category in Method section

b. What are the clinical implications in practice of these findings?

i. Dentist education?

ii. Patient education?

iii. Focus on preventive care?

iv. Increased FDA labeling of products

Reviewer #2: This manuscript addresses an important and timely clinical question: whether long-term opioid therapy (LTOT) is associated with dental disease, in light of recent FDA warnings about buprenorphine. Using a large, national Veterans Affairs (VA) cohort, the authors find a modest but statistically significant association between LTOT and infection-related dental outcomes. The topic is highly relevant to public health and pain management policy. However following problems inherent to the design was found.

1.LTOT initiation is likely correlated with pain severity, comorbidities, and behavioral risk factors (e.g., smoking, diet, poor hygiene) that independently affect oral health. Explicitly acknowledge confounding by indication. Consider propensity-score methods (matching or weighting) or sensitivity analysis for unmeasured confounding.

2. More than half of the cohort lacked dental coverage (“unknown/missing” = 56%), potentially biasing outcome ascertainment. Quantify missingness impact (e.g., sensitivity analysis or multiple imputation). Discuss access-related ascertainment bias more explicitly.

3. Temper causal language. Use “associated with” consistently. Clarify temporality (baseline dental status exclusion, lag analyses).

4. Some sections (Methods, Tables 1–3) are overly dense and difficult to follow. Minor editorial inconsistencies (e.g., “BMI ≥ 309” likely a typo for ≥ 30).

5. VA population is predominantly older White males with unique service-related exposures and dental benefits. Generalizability to women, younger adults, and non-veterans is limited. State this limitation explicitly in the Discussion.

6. The manuscript is clearly written, though at times repetitive. The abstract accurately summarizes findings but should include specific effect sizes and limitations to reflect balance.

7. The “composite infection-related dental outcome” aggregates caries, oral infections, and tooth loss from administrative codes. These conditions differ in etiology and clinical course; combining them risks obscuring mechanism-specific associations. Disaggregate CIDO components where possible (e.g., separate analyses for caries vs. extractions). Discuss diagnostic validity of administrative codes and potential misclassification bias.

Few of the problems are mentioned in the manuscript

6. PLOS authors have the option to publish the peer review history of their article (what does this mean? ). If published, this will include your full peer review and any attached files.

**Do you want your identity to be public for this peer review?** For information about this choice, including consent withdrawal, please see our Privacy Policy .

Reviewer #1: No

Reviewer #2: No

---

## [Author Response · Author response to Decision Letter 1]

26 Dec 2025

We appreciate the reviewers’ thoughtful comments and suggestions to improve our manuscript. Below we respond to each comment. We believe the edits provide a more detailed and balanced presentation of the study and appreciate the opportunity to submit our revised manuscript.

Reviewers' comments:

Comments to the Author

1. Is the manuscript technically sound, and do the data support the conclusions?

Reviewer #1: Yes

Reviewer #2: Partly

2. Has the statistical analysis been performed appropriately and rigorously?

Reviewer #1: Yes

Reviewer #2: Yes

3. Have the authors made all data underlying the findings in their manuscript fully available?

Reviewer #1: Yes

Reviewer #2: Yes

4. Is the manuscript presented in an intelligible fashion and written in standard English?

Reviewer #1: Yes

Reviewer #2: Yes

5. Review Comments to the Author

Reviewer #1:

1. Recommend changing title to “The association of long-term opioid therapy and Composite Infection-related Dental Outcomes”

We have made the suggested change to the manuscript.

2. In Methods, you state primary outcome related to new-onset dental caries. How was this determined given that caries is a chronic bacterial infection that may be present before subjective reports by patients or clinical/radiographic findings?

We have clarified that the outcome is any new diagnosis (i.e., post-baseline) of a CIDO-defining condition (dental caries, oral infections, or tooth loss).

Methods (lines 119-123) now state:

“The primary outcome was the composite infection-related dental outcome (CIDO), a dichotomous indicator of any new (i.e., post-baseline) diagnosis of dental caries, oral infections, or tooth loss according to International Classification of Diseases (ICD) and Current Procedural Terminology codes (adjudicated by dental expert coauthors GG, MH, MJ, KS). (Supplemental Table 1).”

Please define term “oral infections” to assist readership in knowing if this is referring to periodontal/endodontic and/or post-surgical infections. Was tooth loss inclusive to oral infections and exclusive of orthodontic and/or prophylactic removal of third molars?

We appreciate the reviewer’s important point. Codes used to define elements of the composite outcome are included in Supplemental Table 1 and referenced in Methods. We have added descriptions for all codes in the supplemental table. All oral infections and occurrences of tooth loss were included to provide a sensitive measure of the outcome. In this adult population of Veterans with mean age of 58 years, we believe that orthodontic and/or prophylactic removal of third molars represents such a small share of tooth loss that their inclusion is very unlikely to meaningfully influence the results.

3. In Methods, you mention you adjusted Cox model for substance use (drug use disorder, alcohol use disorder)

a. Type of drugs misused may be a confounding variable in this study as methamphetamine and opioids alter salivary function; methamphetamine decreases salivary pH.

We agree that use of other substances may account for CIDO. Although we did not separate out drug type, controlling for substance use in the model allowed us to assess the residual effect of LTOT on CIDO. We now suggest modeling the effects of individual drugs as a direction for future research in the Discussion, lines 221-223:

“Whereas in this study we controlled for substance use disorders, future research should assess the unique risks of specific substance use in the context of LTOT exposure.”

4. In Methods, please describe why patients with cancer, comprehensive dental coverage, and less than 180 days of follow-up were excluded? Please define the requirements for veterans to obtain comprehensive dental coverage? Are they more susceptible to long-term opioid exposure? Why?

All patients were included in the primary analyses. In one sensitivity analysis, we included only patients with at least 180 days of follow-up to standardize the time at risk for event occurrence.

We excluded patients with cancer and, in a separate sensitivity analysis, restricted to patients with comprehensive dental coverage to test for stability of model estimates, because of the potential vulnerability of patients in these subgroups to develop infection.

Dental care eligibility is defined in detail in statutes 38 USC §1710 and §1712,1,2 which we now cite in the paper. Because the statutes are highly detailed with many qualifiers and exceptions, we have elected not to summarize them in the paper to eliminate risk of misrepresentation and avoid diversion from the focus of the paper. Veterans with qualifying conditions for dental coverage may be more susceptible to new dental disease as measured by CIDO. It is also possible that Veterans with comprehensive dental coverage are seen for dental care more often in the VA, influencing detection and recording of CIDO-related events. We attempted to control for this important risk for confounding by matching patients prescribed LTOT to those not prescribed LTOT on the amount of VA dental coverage received (please also see our response to Q7).

We have revised language in the manuscript to clarify the purpose of sensitivity analyses (Methods, lines 144-149):

“We conducted several sensitivity analyses to test for stability in model estimates: (1) excluding patients with cancer who may be uniquely vulnerable to infection, (2) restricting to patients with comprehensive VA dental coverage who may be particularly vulnerable to infection and whose dental care records may be differentially available in the VA EHR, and (3) restricting to patients with ≥180 days of follow-up time to standardize the time at risk for CIDO.”

5. Tables need to be organized with data presented in order mentioned in text (race, smoking status, urban/rural, pain, MH, CV, SUD etc.).

We have reorganized tables and text for consistent ordering of variables.

Too much information is provided in tables. Recommend increasing the number of tables.

We agree the tables were dense with information. Rather than break up tables, we removed non-essential information. Specifically, p-value columns in all tables were replaced with footnotes and the column with unadjusted estimates in Table 2 was removed and the results are instead reported in text.

6. Results:

a. State patients prescribed LTOT live in rural settings-data shows 58.93% urban vs. 39.34% rural with more living in urban settings-please explain and/or correct your result statement

We have clarified that people prescribed LTOT were more likely than those not prescribed LTOT to live in rural settings (Results, lines 164-165).

7. Discussion

a. Why are patients with VA comprehensive dental coverage already extremely vulnerable to dental infection? Please define this coverage category in Method section.

Please see our response to comment #4. We have revised the Methods section to more accurately state that patients were matched on their amount of VA dental coverage (none, less than comprehensive, comprehensive) (line 114).

b. What are the clinical implications in practice of these findings?

i. Dentist education?

ii. Patient education?

iii. Focus on preventive care?

iv. Increased FDA labeling of products

We appreciate the reviewer’s question and agree there are several potential clinical implications, especially if these results are supported by future research. We have included implications for practice in the Discussion, acknowledging the importance of future research to disentangle effects of LTOT from those of other substance use. (lines 208-212).

Reviewer #2: This manuscript addresses an important and timely clinical question: whether long-term opioid therapy (LTOT) is associated with dental disease, in light of recent FDA warnings about buprenorphine. Using a large, national Veterans Affairs (VA) cohort, the authors find a modest but statistically significant association between LTOT and infection-related dental outcomes. The topic is highly relevant to public health and pain management policy. However following problems inherent to the design was found.

1. LTOT initiation is likely correlated with pain severity, comorbidities, and behavioral risk factors (e.g., smoking, diet, poor hygiene) that independently affect oral health. Explicitly acknowledge confounding by indication. Consider propensity-score methods (matching or weighting) or sensitivity analysis for unmeasured confounding.

We appreciate the reviewer’s point and acknowledge as a limitation the risk for residual confounding.

Discussion, lines 212-216, now state:

“However several study limitations should be considered when interpreting this study’s results, including the risk for confounding by indication. Specifically, despite our effort to control for covariates of LTOT prescribing including substance use and medical comorbidities, unmeasured confounders may have driven observed associations attributed to LTOT.”

2. More than half of the cohort lacked dental coverage (“unknown/missing” = 56%), potentially biasing outcome ascertainment. Quantify missingness impact (e.g., sensitivity analysis or multiple imputation). Discuss access-related ascertainment bias more explicitly.

Thank you for this point. As indeed it was the case that more than half of the cohort had undetermined dental coverage, we matched on (known or unknown) dental coverage to account for this to some degree. However, we acknowledge in the Discussion, lines 218-219, that patients without VA dental coverage may have had CIDO recorded in other systems not reviewed in the current study. We now specifically identify this as risk for access-related ascertainment bias.

3. Temper causal language. Use “associated with” consistently. Clarify temporality (baseline dental status exclusion, lag analyses).

We have reviewed the manuscript and tempered causal language. With respect to the temporal relationship between predictors and outcomes, incident LTOT exposure determined the start of the post-baseline period (i.e., index date). Patient characteristics were ascertained during the baseline period (i.e., the 12 months prior to index date; Methods lines 117-118). The outcome, new CIDO infections, were determined post-baseline. Patients with pre-existing (i.e. baseline) CIDO were excluded (Methods lines 105-106). Thus, all predictors preceded the outcome.

We now clarify that CIDO was recorded post-baseline on line 120 and reinforce that patient characteristics were ascertained during baseline on lines 128-29.

4. Some sections (Methods, Tables 1–3) are overly dense and difficult to follow.

We have reduced the information in the tables (please see response to Reviewer 1, #5) and edited the Methods section to reduce sentence length and present information in smaller segments.

Minor editorial inconsistencies (e.g., “BMI ≥ 309” likely a typo for ≥ 30).

The 9 in 309 was a superscripted citation. We regret the misinterpretation and have moved the citation outside of the parentheses. We have reviewed the entire manuscript for errors and inconsistencies.

5. VA population is predominantly older White males with unique service-related exposures and dental benefits. Generalizability to (people outside the VA) women, younger adults, and non-veterans is limited. State this limitation explicitly in the Discussion.

We now name unclear generalizability as a limitation in the Discussion, lines 222-224:

“Finally, given the study’s focus on a patient sample receiving services within the VA, results may not generalize to a broader population of patients.”

6. The manuscript is clearly written, though at times repetitive. The abstract accurately summarizes findings but should include specific effect sizes and limitations to reflect balance. We have added effect estimates associated with the primary variable of interest (LTOT) and limitations to the abstract. We revised the conclusion to reduce repetition with results.

7. The “composite infection-related dental outcome” aggregates caries, oral infections, and tooth loss from administrative codes. These conditions differ in etiology and clinical course; combining them risks obscuring mechanism-specific associations. Disaggregate CIDO components where possible (e.g., separate analyses for caries vs. extractions).

We appreciate this suggestion and agree disaggregation is an important next step for this work, however it is beyond the scope of this initial analysis. We now suggest this as a direction for future research in the Discussion, lines 223-226:

“Further, as conditions comprising CIDO (e.g., oral infection, tooth loss) differ in etiology and clinical course, our use of a composite outcome may mask mechanism-specific associations. Thus, future research on this topic should assess effects of LTOT on individual CIDO conditions.”

Discuss diagnostic validity of administrative codes and potential misclassification bias.

We are not aware of any formal validation studies of administrative codes related to oral health care in the VA. A recent study using Million Veteran Program data assessed correspondence between ICD codes and Veteran self-reported oral health (a measure not vulnerable to bias by differences in recording), showing a strong correlation between ICD-coded periodontitis and tooth loss and Veterans’ self-reported worsened oral health. They also found the percentage of Veterans with ICD-coded tooth loss increased as patients’ self-reported oral health declined.3 The codes used in this study were similar to those used by coauthor Suda and colleagues (2019),4 and were based on review and consensus of coauthors with expertise in VA dental care (GG, MH, MJ, KS). This is now noted in the Methods section, line 122.

Reference

1. United States Code. (2015). Eligibility for hospital, nursing home, and domiciliary care, 38 U.S.C. § 1710. Retrieved from https://uscode.house.gov/view.xhtml?req=granuleid:USC-2015-title38-section1710&num=0&edition=2015

2. United States Code. (2023). Dental care; drugs and medicines for certain disabled veterans; vaccines, 38 U.S.C. § 1712. Retrieved December 26, 2025, from https://www.govinfo.gov/app/details/USCODE-2023-title38/USCODE-2023-title38-partII-chap17-subchapII-sec1712

3. Yu YH, Pridgen KM, Nelson TJ, Miller DR, Wells JM, Assimes TL, O’Donnell CJ, Tsao PS, Chang KM, Lynch JA. Oral health, inflammation, and cardiometabolic factors in the VA million veteran program. JDR Clinical & Translational Research. 2025 Oct;10(4):457-68.

4. Suda KJ, Calip GS, Zhou J, Rowan S, Gross AE, Hershow RC, Perez RI, McGregor JC, Evans CT. Assessment of the appropriateness of antibiotic prescriptions for infection prophylaxis before dental p

---

## [Editor Report · Decision Letter 1]

6 Jan 2026

The association between long-term opioid therapy and Composite Infection-related Dental Outcomes

PONE-D-25-46805R1

Dear Dr. Black,

We’re pleased to inform you that your manuscript has been judged scientifically suitable for publication and will be formally accepted for publication once it meets all outstanding technical requirements.

Kind regards,

Kimberly Page, PhD, MPH

Academic Editor

PLOS One

Additional Editor Comments (optional):

The Authors' responses to critiques and revisions are clear and have improved this manuscript which presents a really excellent analysis of oral disease associated with long-term opioid use. I hope this paper is widely read and cited!
---

## [Editor Report · Acceptance letter]

PONE-D-25-46805R1

PLOS One

Dear Dr. Black,

I'm pleased to inform you that your manuscript has been deemed suitable for publication in PLOS One. Congratulations! Your manuscript is now being handed over to our production team.

Kind regards,

on behalf of

Dr. Kimberly Page

Academic Editor

PLOS One